# In Silico/In Vitro Hit-to-Lead Methodology Yields SMYD3 Inhibitor That Eliminates Unrestrained Proliferation of Breast Carcinoma Cells

**DOI:** 10.3390/ijms21249549

**Published:** 2020-12-15

**Authors:** Ilham M. Alshiraihi, Dillon K. Jarrell, Zeyad Arhouma, Kelly N. Hassell, Jaelyn Montgomery, Alyssa Padilla, Hend M. Ibrahim, Debbie C. Crans, Takamitsu A. Kato, Mark A. Brown

**Affiliations:** 1Cell and Molecular Biology Program, Colorado State University, Fort Collins, CO 80523-1005, USA; alshiraihi@gmail.com (I.M.A.); zkrahuma@rams.colostate.edu (Z.A.); khassell@colostate.edu (K.N.H.); debbie.crans@colostate.edu (D.C.C.); tkato@rams.colostate.edu (T.A.K.); 2Department of Biology, University of Tabuk, Tabuk 47713, Saudi Arabia; 3Department of Bioengineering, University of Colorado Anschutz Medical Campus, Aurora, CO 80045-7109, USA; dillom.jarrel@cuanschutz.edu; 4Department of Chemistry, Colorado State University, Fort Collins, CO 80523-1872, USA; 5Department of Biomedical Sciences, Colorado State University, Fort Collins, CO 80523-1617, USA; Jaelyn.Montgomery@rams.colostate.edu (J.M.); aepad20@rams.colostate.edu (A.P.); 6Department of Medical Biochemistry, Zagazig University, Zagazig 44511, Egypt; hendibrahim1@gmail.com; 7Department of Environmental & Radiological Health Sciences, Colorado State University, Fort Collins, CO 80523-1618, USA; 8Department of Clinical Sciences, Colorado State University, Fort Collins, CO 80523-1678, USA; 9Epidemiology Section, Colorado School of Public Health, Fort Collins, CO 80523-1612, USA; 10Institute for Learning and Teaching, Colorado State University, Fort Collins, CO 80523-1052, USA; 11Department of Ethnic Studies, Colorado State University, Fort Collins, CO 80523-1790, USA

**Keywords:** hit-to-lead, in silico drug development, SMYD3, methyltransferase, Inhibitor-4, breast cancer, cell proliferation, cell cycle, apoptosis

## Abstract

SMYD3 is a lysine methyltransferase that regulates the expression of over 80 genes and is required for the uncontrolled proliferation of most breast, colorectal, and hepatocellular carcinomas. The elimination of SMYD3 restores normal expression patterns of these genes and halts aberrant cell proliferation, making it a promising target for small molecule inhibition. In this study, we sought to establish a proof of concept for our in silico/in vitro hit-to-lead enzyme inhibitor development platform and to identify a lead small molecule candidate for SMYD3 inhibition. We used Schrodinger^®^ software to screen libraries of small molecules in silico and the five compounds with the greatest predicted binding affinity within the SMYD3 binding pocket were purchased and assessed in vitro in direct binding assays and in breast cancer cell lines. We have confirmed the ability of one of these inhibitors, Inhibitor-4, to restore normal rates of cell proliferation, arrest the cell cycle, and induce apoptosis in breast cancer cells without affecting wildtype cell behavior. Our results provide a proof of concept for this fast and affordable small molecule hit-to-lead methodology as well as a promising candidate small molecule SMYD3 inhibitor for the treatment of human cancer.

## 1. Introduction

SMYD (SET and MYND domain-containing) protein family members constitute a class of methyltransferases that regulate a wide range of normal cellular processes [1,2,3] and are also involved in several tumorigenic pathways [3,4,5]. SMYD3, the third member of the SMYD family, transfers methyl groups to lysine 4 on histone H3 (H3K4) and lysine 5 on histone 4 (H4K5), a residue that was previously thought to only undergo acetylation [1,2,3]. Overexpression of SMYD3 results in increased cell proliferation and activates many genes associated with cancer cell transformation [6] and metastasis [7]. Several studies have revealed that lung, breast, pancreatic, colorectal, and hepatocellular carcinoma are highly associated with SMYD3 overexpression [3,8,9]. In vitro studies using NIH3T3 cells have demonstrated that SMYD3 involvement in uncontrolled proliferation is one of the crucial stages in tumorigenesis. Furthermore, the growth of breast, hepatocellular, and colorectal carcinoma cell lines have been impaired significantly through SMYD3 knockdown [3,8]. These studies demonstrate that the oncogenic impact of SMYD3 is mediated in part by its histone methylation activity and the resulting impact on the expression of oncogenes. These downstream genes include NKX2.8 [3], WNT10B [8], TERT [10], cMET [11], and CDK2 [12]. SMYD3 is also known to regulate cancer cell proliferation and viability through its interaction with vascular endothelial growth factor receptor 1 (VEGFR1) [13] and estrogen receptor (ER) [14], both of which are non-histone proteins. The role of SMYD3 in ER-mediated transcription through its histone methyltransferase activity is not fully known. SMYD3 acts as a cofactor of ERα and promotes its efficacy in response to bound ligands. In addition, SMYD3 interacts with ER in the ligand binding domain and activates the transcriptional machinery of downstream genes [14]. Collectively, these studies indicate SMYD3 as a potential therapeutic target for cancer treatment.

The small molecule drug development process is notoriously expensive, time consuming, and inefficient. After target identification, identifying hit compounds with significant activity requires enormous small-molecule libraries and hours of experimentation. Optimizing hit compounds to identify leads that have activity in cells and that meet initial in vitro toxicity criteria often requires several rounds of iteration and molecular synthesis. Because of the difficulty and cost of this process, scientific literature is saturated with studies that identify proteins that are “promising therapeutic targets” but that proceed no further towards actual therapeutic development. In this study, we sought to establish a simple and affordable hit-to-lead methodology that could be implemented by average research laboratories that have elucidated druggable proteins. Using in silico screening to identify initial hits and restricting the initial library to purchasable compounds, we implemented the Small Molecule Drug Discovery Suite (Schrodinger, Inc., NY, USA) to predict the binding affinity of a library of 137,990 molecules [15,16,17] and thus have demonstrated the ability to identify lead small molecule inhibitors without the need for physical compound libraries or in-house chemical synthesis.

Using SMYD3 as a target protein, we implemented our screening methodology and report a novel small molecule SMYD3 inhibitor (Inhibitor-4) that impairs breast cancer cell proliferation without affecting normal cells, illustrating the potential of SMYD3 inhibitors in the clinical management of breast cancer as well as a proof of concept for this drug development platform. We used two breast cancer cell lines (MCF7 and MDA-MB-231) that were previously shown to overexpress SMYD3 [8,14,18,19,20] compared with the wild type breast epithelial cell line MCF10A [18,19] (Table 1). After initial hit identification in vitro, we purchased and tested five novel small molecule SMYD3 inhibitors and discovered that Inhibitor-4 significantly reduces breast cancer proliferation, arrests the cell cycle, and induces apoptosis without impacting wild type cells. In all experiments, we used a previously-identified SMYD3 small molecule inhibitor, BCI-121, as a positive control [21].

## 2. Results

### 2.1. Inhibitor-4 Decreases SMYD3-Mediated H3 Methylation

After our iterative in silico screening using Schrodinger software (Glide^®^, Maestro^®^, LigPrep^®^, and Epik^®^), we purchased the top five hit compounds for testing. Hits were defined as the drug-like small molecules with the lowest free binding energy when docked in the protein-target binding pocket of SMYD3. The predicted free binding energies of the five lead compounds ranged from −7.2 kJ/mol to −9.1 kJ/mol, compared to the natural protein ligand’s predicted free binding energy of only around −1 kJ/mol (fragment of VEGFR1). We used an in vitro methylation assay using purified Histone 3 (H3) to assess the ability of the five lead in silico-designed SMYD3 inhibitor candidates to decrease SMYD3 enzymatic activity. We demonstrated that Compound 4 (Inhibitor-4) significantly reduces SMYD3-mediated Histone 3 methylation (70% reduction), while the other novel compounds did not show significant differences. H3 was chosen because of previous studies that demonstrated SMYD3 methylates H3 preferentially (Figure 1) [3].

### 2.2. Inhibitor-4 and BCl-121 are Stable in d_6_-DMSO Solution

Because of the limited solubility of selected molecules in aqueous solution and in media, we dissolved BCI-121 and Inhibitor-4 in d_6_-DMSO solution to record and analyze the 1D ^1^H NMR spectra of both compounds. The major species attributed to Inhibitor-4 and BCI-121 were observed at time 0 and 24 h, as shown in Appendix A (BCI-121) and Appendix A (Inhibitor-4). The ^1^H NMR peaks of the fresh and aged samples for Inhibitor-4 showed no observable difference in the presence of the major component (67%) and minor component (33%) peaks as a function of time, suggesting that no hydrolysis is taking place during the experiment for Inhibitor-4. For BCI-121, 70% of the major species was present at time 0, however after 24 h this decreased slightly to 68%, suggesting that the positive control may be slightly less stable than Inhibitor-4.

### 2.3. SMYD3 Is Overexpressed in Breast Cancer Cells

Western blot and immunocytochemistry were carried out to test the expression levels of SMYD3 using anti-SMYD3 antibody in normal and breast cancer cell lines. Western blot data have indicated that SMYD3 was highly expressed in breast cancer cell lines (1.8-fold in MCF7 and 2.6-fold in MDA-MB-231) compared to normal cell line (Figure 2).

Additionally, immunocytochemistry data have shown elevated levels of SMYD3 expression in breast cancer cell lines comparing to normal cell line (Figure 2a,b). Therefore, increased SMYD3 expression could be correlated with breast carcinogenesis.

### 2.4. Inhibitor-4 Inhibits Growth of Breast Cancer Cells

The impact of SMYD3 inhibitors on growth of breast cancer cells was tested by adding 50, 100 and 200 μM of Inhibitor-4 or BCI-121 to breast cancer cell lines (MCF7 and MDA-MB-231) and normal breast epithelial cell line (MCF10A). The number of cells was determined daily and the population doubling times were quantified (Figure 3). For MCF7 (breast cancer) cells, the basal doubling time for MCF7 was 38 h, while 40 h for MDA-MB-231. Using the positive control inhibitor, a concentration of 200 μM caused approximately 2-fold suppression of MCF7 cellular growth (Figure 3a). Using Inhibitor-4, however, a clear dose-dependent suppression in growth was observed with the first significant reduction observed at a concentration of 50 μM (Figure 3b). In the MDA-MB-231 cell line, a significant delay in the cellular growth was observed with 200 μM BCI-121 and only 50 μM Inhibitor-4 (Figure 3c,d).

For MCF10A (normal) cells, the effect of the SMYD3 inhibitors was limited. The basal doubling time for MCF10A was 28 h. Interestingly, no delay was noticed with 50, or 100 μM concentrations of either inhibitor. Treatment of the normal cells with 200 μM of Inhibitor-4 resulted in a minor, not significant, growth delay (approximately 5%), while treatment with 200 μM BCI-121 resulted in a major growth delay (Figure 3e,f). These results suggest that Inhibitor-4 shows more growth inhibition than BCI-121 and causes significant inhibition in cancer cell growth while only modestly impacting healthy cells.

### 2.5. Inhibitor-4 Suppresses Breast Cancer Cell Colony Formation

To determine the effects of Inhibitor-4 on the colony formation of breast cancer cells and normal cell lines, the cells were treated with various concentrations of Inhibitor-4 and BCI-121 (10, 50, 100, 150 and 200 μM) and incubated for 2 weeks. Treatment with Inhibitor-4 significantly suppressed clonogenic activity in MCF7 and MDAMB-231 cells at concentrations of 50, 100, 150 and 200 μM (Appendix A) compared to normal MCF10A cell line (Appendix A). Similarly, BCI-121 suppressed colony formation on MCF7 at nearly all concentrations (Appendix A) and MDA-MB-231 cells at 150 and 200 μM concentrations (Appendix A). Surprisingly, a significant decrease in colony formation of MCF10A (normal) cells was also observed at 200 μM concentration of BCI-121 (Appendix A) compared to Inhibitor 4, which did not affect MCF10A survival (Appendix A). This result again suggests the improved inhibition effect of Inhibitor-4 compared to BCI-121.

### 2.6. Inhibitor-4 Reduces Cell Viability in MCF7 Cells

The effect of Inhibitor-4 on the viability of wild type and cancer cell lines was evaluated using an MTT assay at different time points (24, 48, 72, and 96 h). Cells were treated with the vehicle (DMSO 0.1%, 0.15% and 0.2%), BCI-121, or Inhibitor-4 (Appendix A). Treatment with BCI-121 caused significant decreases in cell viability in both breast cancer (MCF7 and MDA-MB-231) and wild type (MCF10A) cell lines at multiple time points, particularly at concentrations of 150 and 200 μM (Appendix A). However, Inhibitor-4 caused significant decreases in cell viability only in the cancer cell lines (MCF7 at 150 and 200 μM, MDA-MB-231 at 200 μM; Appendix A). No concentration of Inhibitor-4 impacted MCF10A cellular viability (Appendix A). Collectively, these data suggest that Inhibitor-4 is a promising, cancer-specific inhibitor that reduces cancer cell line viability and growth without affecting normal cells.

### 2.7. Inhibitor-4 Induces Cell Cycle Arrest in Breast Cancer Cells

To investigate whether the growth inhibitory effect of Inhibitor-4 on breast cancer cells was due to cell cycle arrest, we conducted cell cycle analysis using Propidium Iodide (PI) staining. Cells were treated with 200 μM of BCI-121 or Inhibitor-4 for 24 h. As shown in Figure 4 and Appendix A, treatments with both the positive control inhibitor and Inhibitor-4 induced G1 arrest and reduced S phase in MCF7 cells (Appendix A and Figure 4a,b) compared to MCF7 control (Appendix A). Also, both treatments led to G1 arrest in MDA-MB-231 (Appendix A and Figure 4c,d) compared to MDA-MB-231 control (Appendix A). Therefore, BCI-121 and Inhibitor-4 prompted an increase in G1 fractions. However, treatments with BCI-121 and Inhibitor-4 did not induce cell cycle arrest (Appendix A) compared to control (Appendix A) or show statistical differences in normal MCF10A cells (Figure 4e,f).

### 2.8. Inhibitor-4 Promotes Apoptosis in Breast Cancer Cells

To reveal whether Inhibitor-4 induce apoptosis on breast cancer cell line or not, we performed apoptosis assay using APC Annexin V/PI followed by flow cytometry analysis. After 48 h of Inhibitor-4 treatment, the percentage of live cells decreased to 71% in both breast cancer cell lines (from 91% in MCF7 and 95% in MDA-MB-231) as demonstrated by flow cytometry (Appendix A and Figure 5b,d). Also, treatment with Inhibitor-4 showed increase in late apoptosis and necrosis percentages in MCF7 (Appendix A and Figure 5b), while MDA-MB-231 showed early apoptosis with treatment of Inhibitor-4 (Appendix A and Figure 5d). BCI-121 caused late apoptosis in MCF7 and both early and late apoptosis in MDA-MB-231 cells, in addition to necrosis in MCF7 cells (Appendix A and Figure 5a,c). Neither treatment caused significant differences in apoptosis nor necrosis in MCF10A cells (Appendix A and Figure 5e,f).

Apoptosis induction through SMYD3 inhibitors was also tested using Caspase-3/7 activity assay. The data have shown increases in Caspase-3/7 activity in MDA-MB-231, however, no significant differences in MCF7, which is Caspase-3/7 independent apoptosis pathway, and MCF10-A.

## 3. Discussion

Aberrant expression of SMYD3 has been shown to be oncogenic and is essential for the proliferation of most colorectal, hepatocellular, and breast carcinomas, as well as prostate cancer [3,8]. Over 80 genes (including highly regulated homeobox genes, cell cycle regulators, and oncogenes) display altered expression because of aberrant upregulation of SMYD proteins [1,2,3]. Specifically, SMYD3 over-expression is highly associated with cancer development by regulating tumor proliferation, metastasis, invasion, and apoptosis [22]. Several studies have shown that SMYD3 regulates the oncogenic RAS signaling pathway by integrating a cytoplasmic-kinase signaling cascade, resulting in accelerated cell proliferation and differentiation [9]. Another study demonstrated that SMYD3 is essential for estrogen receptor-mediated transcription in breast cancer cells by down-regulating SMYD3 via RNA interference [14]. SMYD3 mediated-H2A.Z methylation has also been shown to trigger cyclin A1 gene expression, leading to cell cycle activation in breast cancer cells [23]. Knock down of SMYD3 in ovarian cancer tissues leads to upregulation of CDKN2B (p15INK4B), CDKN2A (p16INK4), CDC25A and CDKN3 as members of cyclin-dependent kinase inhibitors (CDK) [24]. Inducing apoptosis via silencing of SMYD3 has also been observed in ovarian cancer in vivo and has been accredited to the upregulation of CD40LG and downregulation of BIRC3 [24]. BIRC3 is a member of the inhibitors of apoptosis proteins (IAP) family and relates to many cancers in cases of aberrant overexpression because it can prevent apoptotic signals [25,26]. Therefore, it is likely that SMYD3 inhibitors can trigger apoptosis by down-regulating BIRC3. In addition, another study demonstrated that the MCF7 cell line lacks Caspase 3, which is essential for apoptosis, however in the absence of Caspase 3, Caspase 6 can be activated as an alternative mechanism to trigger apoptosis. As a result, under cellular stress, MCF7 cells undergo apoptosis in response to Caspase 6 and necrosis in response to TNF-α stimulation [27,28,29]. Despite the connection between SMYD3-overexprssion and several types of carcinogenesis, few studies have targeted SMYD3 inhibition in the context of breast cancer through the design of the inhibitors.

In this study, we sought to design small molecule inhibitors for the inhibition of SMYD3-mediated methylation (Figure 1), proliferation (Figure 3), colony formation (Appendix A), and viability (Appendix A) in breast cancer cells. Specifically, we demonstrated that in silico enzyme models can predict effective competitive enzyme inhibitors by screening vast molecular libraries and predicting binding energies. This approach to small molecule design significantly reduces the time, expense, and equipment that have been required for traditional benchtop small molecule screening until now.

Using Schrodinger^®^ software and several in vitro assays, we demonstrated that one of the hit compounds identified in silico (Inhibitor-4) was able to reduce breast cancer cellular growth and viability without affecting normal breast epithelial cells. In vitro, Inhibitor-4 was shown to inhibit SMYD3-mediated histone methylation. In breast cancer cells, Inhibitor-4 extended cell doubling time (Figure 3). We also demonstrated that Inhibitor-4 arrests the cell cycle in breast cancer cells without affecting normal cells (Figure 4 and Appendix A), which demonstrates an improvement over BCl-121, a previously-developed SMYD3 inhibitor. Finally, the novel SMYD3 inhibitor presented here caused apoptosis in breast cancer cell lines without affecting the normal breast cell line (Figure 5 and Appendix A). However, testing the in vivo SMYD3 specificity of Inhibitor-4 needs to investigate the impacts of Inhibitor-4 on SMYD3-knockdown cells. This could be performed for future characterization and validation.

## 4. Conclusions

In conclusion, we have successfully used in silico compound screening to identify a small molecule that inhibit SMYD3 activity in vitro and reduced cancer cell growth and proliferation. Future work will involve the application of this approach to other therapeutic targets and the continued development and optimization of therapeutics for SMYD3-related cancers.

## 5. Materials and Methods

### 5.1. In Silico Screening

We implemented the Small Molecule Drug Discovery Suite (Schrodinger, Inc., New York, NY, USA) to predict the binding affinity of a library of 137,990 molecules [15,16,17]. This library of molecules was downloaded from the free ZINC15 database, and included all “purchasable” molecules with reported or predicted activity in vitro [30]. The 3D structure of SMYD3 used for in silico docking was uploaded from the Protein Data Bank (PDB) under PDB identification code 5EX3 [31]. After an initial simulation which docked each molecule into SMYD3′s protein-target binding pocket (not its s-adenosylmethionine binding pocket), the top ten hits (most-negative binding energy) were entered into the ZINC15 molecular similarity search engine, and the 50 most-similar compounds to each of the ten leading candidates were again scored using the Schrodinger software (500 total compounds). From this iteration, the top five compounds were purchased and assessed in vitro using SMYD3 methyltransferase assays. After initial experiments, Inhibitor-4 was found to be the most promising and, consequently, it advanced to the cell line experiments described below.

### 5.2. Chemical Compounds

All screened compounds were dissolved in dimethyl sulfoxide (DMSO) as 5, 10 or 100 mM stock solutions. The positive control, BCI-121, is a previously-reported SMYD3 inhibitor shown to reduce the cellular proliferation of colorectal and ovarian cancer [21,24]. It was purchased from Millipore Sigma (1817, Burlington, MA, USA) and dissolved in DMSO at 10 and 100 mM. BCI-121 was used in all experiments to investigate its impacts against breast cancer cell lines and as a positive control inhibitor. All compounds were stored at −20 °C until used for the experiments. Some d_6_-dimethyl sulfoxide (D, 99.9%) containing 0.05% *v*/*v* TMS were used for the ^1^H NMR stability study and the d_6_-dimethyl sulfoxide was purchased from Cambridge Isotope Laboratories, Inc. and used as is. The stock solutions of 10.0 mM of BCI-121 and Inhibitor-4 were prepared immediately before use in d_6_-DMSO.

### 5.3. In Vitro Methylation Assay

In vitro methylation was investigated using a colorimetric assay (BioVision, K986-100, Milpitas, CA, USA). SMYD3 inhibitors (160 nM) were incubated with H3 recombinant protein (1.6 μM; Sigma-Aldrich, SRP0177, St. Louis, MO, USA) for 10 min at room temperature. Next, SMYD3 recombinant protein (100 nM; Sigma-Aldrich, SRP0153, St. Louis, MO, USA) and s-adenosylmethionine (SAM) cofactor (500 μM, methyl donor ligand) were added to the SMYD3 inhibitor and H3 solution in the methylation buffer that was provided with the kit. The absorbance was read using a microplate reader (BioTek, Cytation 5, Winooski, VT, USA) at 570 nm in kinetic mode every 30 s at 37 °C for 45 min. The optical density (OD) of the inhibitors was normalized to the optical density (OD) of the control [3].

### 5.4. NMR Spectroscopy Analysis

The 10.0 mM stock solutions of Inhibitor-4 and BCI-121 were prepared freshly in d_6_-DMSO containing 0.05% *v*/*v* tetramethylsilane (TMS) and diluted to the final concentration of 5.0 mM in d_6_-DMSO. The stabilities of Inhibitor-4 and BCI-121 were determined by 1D ^1^H Nuclear Magnetic Resonance (NMR) spectroscopy on a Bruker 400 MHz NMR spectrometer at 25 °C using routine parameters [32]. 2D-NMR experiment will run to confirm the assignments (data not included) [32]. Chemical shifts were measured against TMS (0 ppm) as an internal reference. The spectra of Inhibitor-4 and BCI-121 were recorded at 0 and 24 h. The spectra were worked up and integrated using Mnova V.14 (MestreLab Research SL, Escondido, CA, USA). The signals in the aromatic region were used to measure the ratio of starting material and hydrolysis product.

### 5.5. Cell Culture

All cells were purchased from ATCC and cultured according to manufacturer recommendations. The human breast epithelial cell line MCF10A (ATCC CRL-10317, Manassas, VA, USA) was used as a normal cell line and grown in DMEM/F12 (Invitrogen, 11330-032, Carlsbad, CA, USA) supplemented with 5% horse serum (Invitrogen, 16050-122, Carlsbad, CA, USA), 1% antibiotic/antimycotic mixture (Millipore Sigma, A5955, Burlington, MA, USA), insulin (10 μg/mL), EGF (20 ng/mL), cholera toxin (100 ng/mL), and hydrocortisone (500 ng/mL) [33]. The mammary gland breast cancer lines MCF7 and MDA-MB-231 (ATCC HTB-22 and 26, Manassas, VA, USA) were grown in DMEM media (Corning, 29818003, Corning, NY, USA) with 10% FBS (Atlas Biologicals, F-0500-A, Fort Collins, CO, USA) and 1% antibiotic/antimycotic mixture (Millipore Sigma, A5955, Burlington, MA, USA) [34]. All cell lines were grown in a humidified incubator at 5% CO_2_ and 37 °C with regular passaging to avoid confluence.

### 5.6. Protein Extraction and Immunoblotting

Total protein was extracted from frozen cells using RIPA buffer (150 mM NaCl, 5 mM EDTA, 50 Tris-HCl pH 8.0, 1% NP-40, 0.5% Na-catecholate and 0.1% SDS) supplemented with protease inhibitor (PI 87785, Life tech, Carlsbad, CA, USA). Protein concentration was determined according to Bradford (1976) using bovine serum albumin as a standard [35]. Fifty micrograms of total protein were separated by SDS-PAGE, transferred to nitrocellulose membrane by electroblotting as described by [36] and probed with the antibodies specific for the indicated proteins [36]. Actin was used as an internal control for normalization. Antibodies for immunoblot detection of SMYD3 (Rabbit monoclonal antibody to SMYD3, ab183498, Abcam, Cambridge, MA, USA) and β-Actin (A5316-100 UL, Sigma-Aldrich, St. Louis, MO, USA) have been used as the primary antibodies. Bound antibodies on blots were detected by HRP--conjugated secondary antibodies (ab205718, Abcam, Cambridge, MA, USA). Detection was done using Clarity Western ECL Substrate (Bio-Rad, Hercules, MA, USA) and visualized using Image Lab Software (Bio-Rad). Densitometric evaluation was performed by ImageJ software (Version 2.0.0).

### 5.7. Immunocytochemistry

Exponentially growing cells were fixed with 4% paraformaldehyde for 15 min at room temperature. After PBS wash, cells were permeabilized with 0.1% SDS, 0.5% Triton X-100 in PBS for 10 min at room temperature. 10% goat serum in PBS was used for blocking for 30 min at room temperature. Primary antibody was diluted in 1:300 and treated for 1 h at 37 °C. Secondary antibody (Alexa488 conjugated anti-rabbit IgG) was diluted for 1:500 and treated for 30 min at 37 °C. DAPI in Vectashield antifade solution was used for mounting. The images were capture by Zeiss Axiophot microscope with Qimaging Exi Aqua camera with Qcapture pro software. Blue signal was obtained with 50 millisecond exposure. Green signal was obtained with 200 millisecond exposure. Signals were quantified by densitometry using ImageJ2 software (Version 2.0.0).

### 5.8. In Vitro Cell Growth Inhibition Assay

Normal and breast cell lines were plated at a density of 20,000 cells/well onto a 6-well plate with different concentrations of Inhibitor-4 and BCI-121. After trypsinization, cell numbers were counted and scored as the number of proliferating cells after treatments at different time points (24, 48, 72 and 96 h) using a Coulter Counter Z2 (Beckman-Coulter Z2 Coulter Particle Count Counter and Size Analyzer, Brea, CA, USA). Data were analyzed and cell doubling time was calculated using GraphPad Prism 6 software (Graph Pad Software, La Jolla, CA, USA) through the exponential growing equation using the exponential growing stage [37].

### 5.9. Clonogenic Cell Survival

A colony formation assay was used to determine cell sensitivity to SMYD3 inhibitors. The self-renewal and proliferative capacities of cells were measured. To form colonies, cells were seeded onto 6 well plates and were treated with varying concentrations of BCI-121 and Inhibitor-4. The plated cells were incubated in a humidified incubator at 5% CO_2_ and 37 °C for two weeks. Then, colonies were fixed with 100% ethanol and allowed to dry for 20 min at room temperature before staining. Colonies were stained using 0.1% crystal violet and allowed to dry before counting. Reproductively viable surviving cells were counted based on the microscopic colonies containing more than 50 cells. From the cell survival fraction, survival curves were drawn using Graph Pad Prism 6 software (Graph Pad Software, La Jolla, CA, USA). At least three independent experiments for each cell line were conducted [38].

### 5.10. MTT Assay

Cells were plated at a density of 5000 cells/well in 96 well plates. After seeding, cells were treated with the vehicle (DMSO 0.1%, 0.15% and 0.2% *v*/*v*) or various concentrations of the screening inhibitors. The plated cells were incubated in CO_2_ and treated for 24, 48, 72 and 96 h. Then, 10 μL MTT solutions (5 mg/mL) were added to each well followed by a 4 h incubation in CO_2_ in the dark. Formazan crystals formed were dissolved in 100 μL of SDS followed by a second 4 h incubation in CO_2_. The absorbance was read using a microplate reader (BioTek Instrument, Cytation 5, Winooski, VT, USA). The optical density (OD) of each sample was subtracted from the optical density (OD) of the background and the Formazan standard curve was determined. Cellular viability of all samples was calculated using the ratio of the inhibitor treated-groups versus vehicle-treated group. Graph bars were obtained using GraphPad Prism 6 software (Graph Pad Software, La Jolla, CA, USA) [39].

### 5.11. Cell Cycle Assay

Cell cycle distributions were analyzed using PI flow cytometry. Cells were plated at density of 5 × 10^5^ cells per well onto 6-well plate. Cells were treated with 200 μM of Inhibitor-4 or BCI-121 and incubated in a humidified incubator at 5% CO_2_ and 37 °C for 24 h. Following incubation, detached cells were collected, washed two times with phosphate buffered saline (PBS). Then, cells were fixed with 70% ethanol in PBS overnight at 4 °C. The fixed cells were washed with PBS twice to remove ethanol thoroughly. The cells were resuspended in propidium iodide staining solution consisting of 20 μg/mL propidium iodide and 200 μg/mL RNase in 0.1% Triton X-100. The stained cells were incubated for 15 min in an incubator at 37 °C. DNA contents were measured subsequently using CyAn ADP analyzer flow cytometry (Beckman Coulter, Fort Collins, CO, USA). Each cell line was gated at 10,000 events and the cell cycle distributions were determined using FLOWJO 10.6 software (FlowJo LLC, Ashland, OR, USA) [40].

### 5.12. Apoptosis Assays

Cell apoptosis was detected using Annexin V, which binds to translocated phosphatidylserine (PS) in the plasma membrane as previously described [41]. Necrosis and late apoptosis were detected using PI to test loss of cell membrane integrity. Briefly, cells were plated and treated with 200 μM of either BCI-121 or Inhibitor-4 for 48 h. Then, the cells were washed with PBS, trypsinized, pelleted, and resuspended in Annexin binding buffer. The cells were stained first with APC Annexin V for 15 min and then with 2.5 μL of PI. The cell mixture was analyzed using a Cytek 4-laser Aurora instrument (Cytek, Fremont, CA, USA). From each sample, a minimum of 3 × 10^4^ events was collected. SpectroFlo software (Cytek, Fremont, CA, USA) was used to analyze the multivariate data. APC Annexin V+/PI+, APC Annexin V-/PI-, APC Annexin V+/PI- or APC Annexin V-/PI+ represented late apoptotic cells, viable cells (intact), early apoptotic cells or necrosis, respectively [42].

In addition to, apoptosis induction by SMYD3 treatments was also assessed using Caspase 3/7 activation. Exponentially growing cells were treated with 200 μL of BC1-121 and Inhibitor-4. After 48 h of incubation, the early apoptosis was measured with the activation of Caspase 3/7 by Caspase-Glo 3/7 kit (Promega, Madison, WI, USA). Glow luminesce of 15,000 cells was measured by Lumat LB9507 (Berthold technologies, Oak Ridge, TN, USA).

### 5.13. Statistical Analysis

The statistical significance of the results in this study was analyzed using GraphPad Prism 6 software (Graph Pad Software, La Jolla, CA, USA) for two-way ANOVA analysis. *p* value of less than 0.05 were considered statistically significant for all analyses.

## Figures and Tables

**Figure 1 ijms-21-09549-f001:**
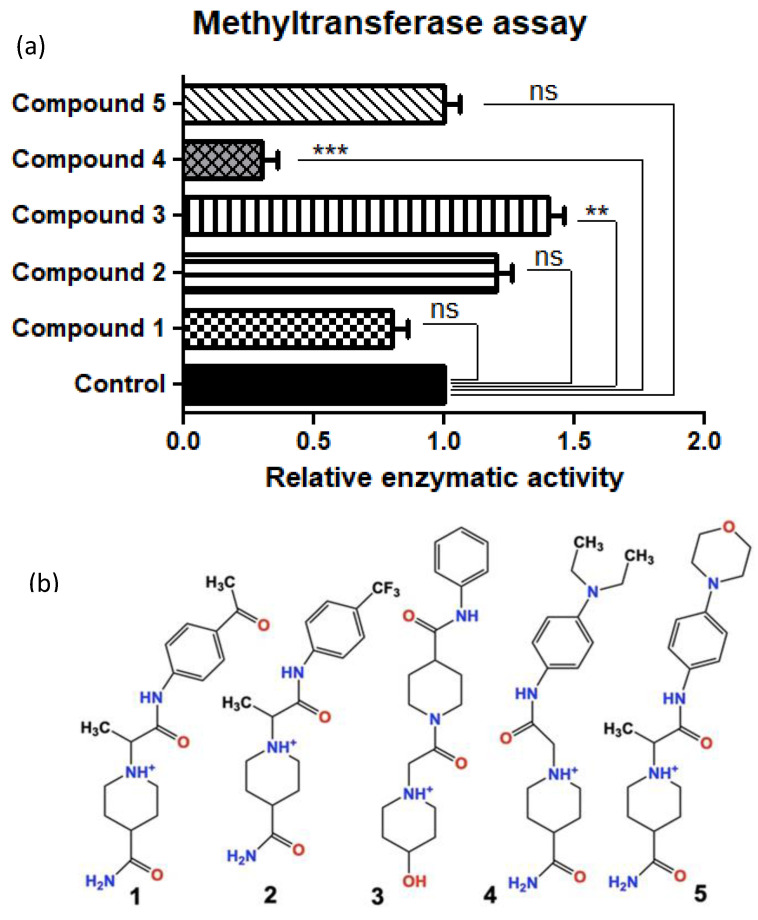
(**a**) Relative SMYD3 activity (top) with the top 5 candidates from in silico testing (bottom) using an in vitro methyltransferase assay (Colorimetric assay). (**b**) Compounds 1–5 drawn in ChemDraw. Compound 4 is from here on referred to as Inhibitor- 4. Error bars display standard error of means. Statistically significant differences from control are indicated by ** *p* < 0.01, *** *p* < 0.001 or ns *p* > 0.05.

**Figure 2 ijms-21-09549-f002:**
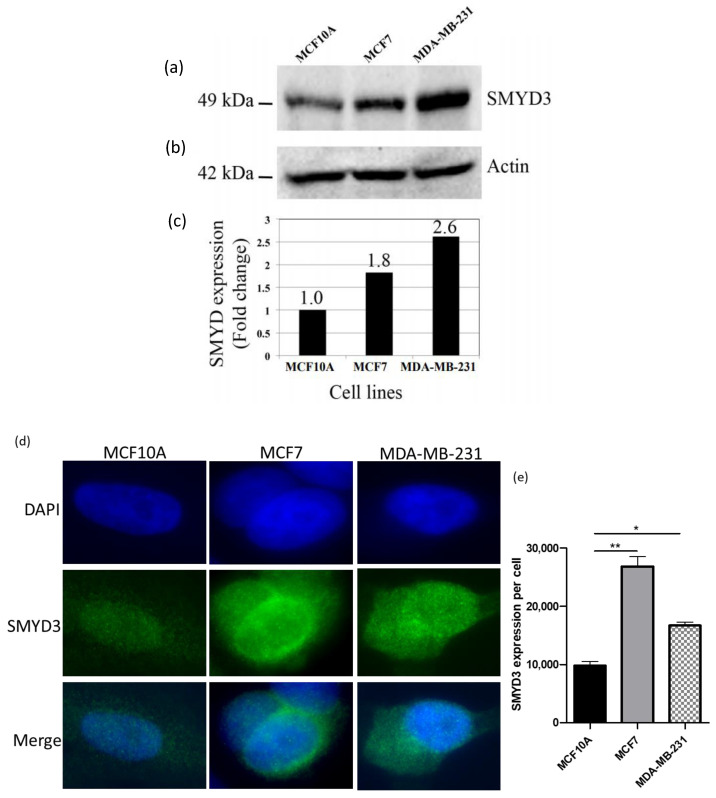
SMYD3 expression using western blot and immunocytochemistry: (**a**) Expression of SMYD3 protein in human cell lines using Western blot. (**b**) Expression of actin served as a quantitative control. (**c**) Western blot analysis shows fold change in SMYD3 expression in the cell lines. (**d**) Expression of SMYD3 protein using immunocytochemistry. (**e**) immunocytochemistry analysis shows SMYD3 intensity in the cell lines. Values are mean ± standard error of the means. Statistically significant differences from control are indicated by * *p* < 0.05, ** *p* < 0.01.

**Figure 3 ijms-21-09549-f003:**
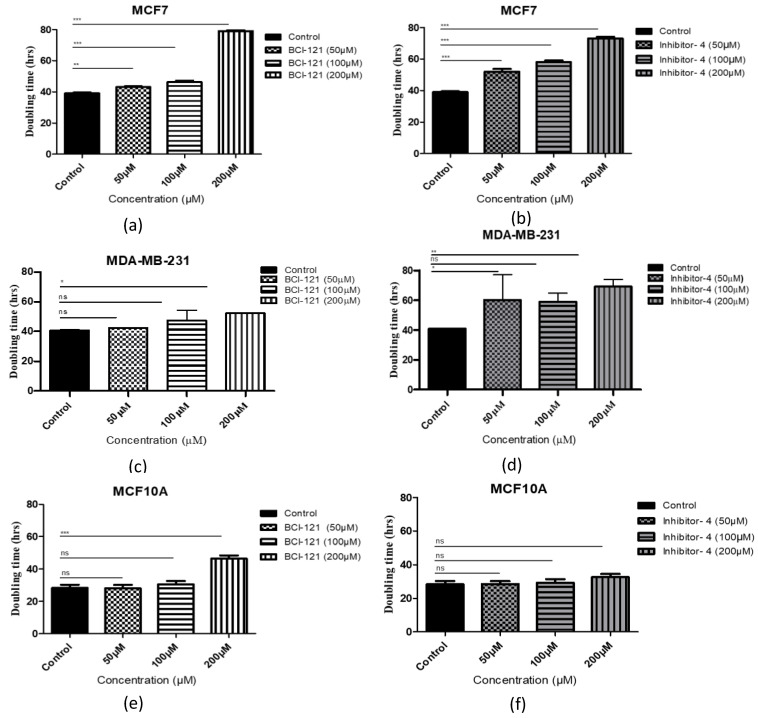
Cell population doubling time with SMYD3 inhibitor treatment. (**a**,**c**,**e**) Cells with BCI-121 as a positive control inhibitor. (**b**,**d**,**f**) Cells with SMYD3 Inhibitor-4. Values are mean ± standard error of the means. Statistically significant differences from control are indicated by * *p* < 0.05, ** *p* < 0.01, *** *p* < 0.001 or ns *p* > 0.05.

**Figure 4 ijms-21-09549-f004:**
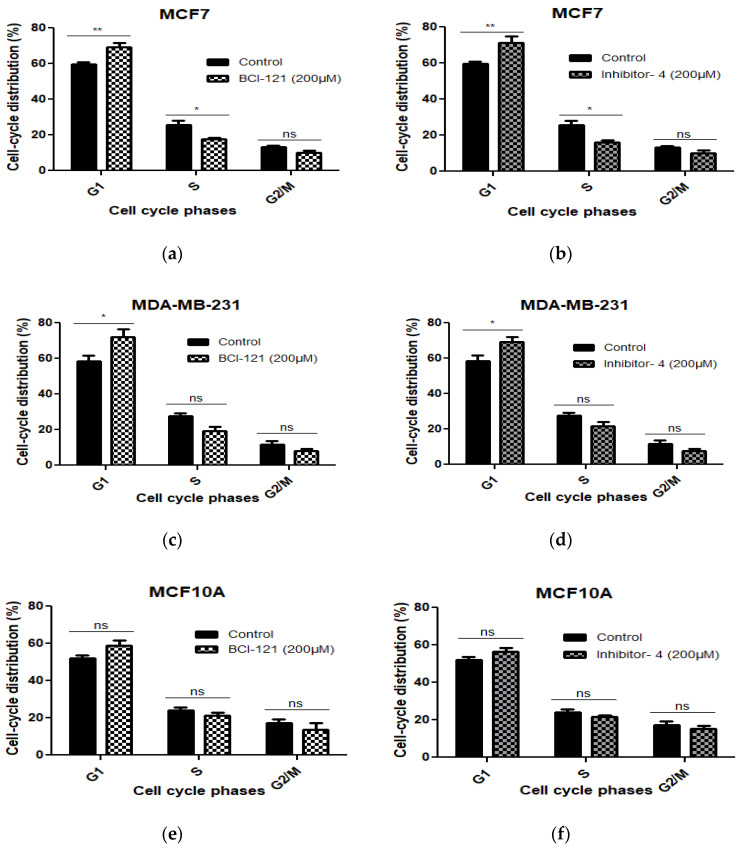
The cell cycle distribution was assessed using PI in MCF7, MDA-MB-231 (breast cancer cell lines) and MCF10A cells (normal breast epithelial cell line) with SMYD3 inhibitor treatments for 24 h and was investigated by flow cytometry. (**a**,**c**,**e**) indicate statistical significant differences on cell cycle phases of the three cell lines treated with BCI-121. (**b**,**d**,**f**) show statistically significant differences on cell cycle phases of the three cell lines treated with Inhibitor-4. Values are mean ± standard error of the means. Statistically significant differences from control are indicated by * *p* < 0.05, ** *p* < 0.01 or ns *p* > 0.05.

**Figure 5 ijms-21-09549-f005:**
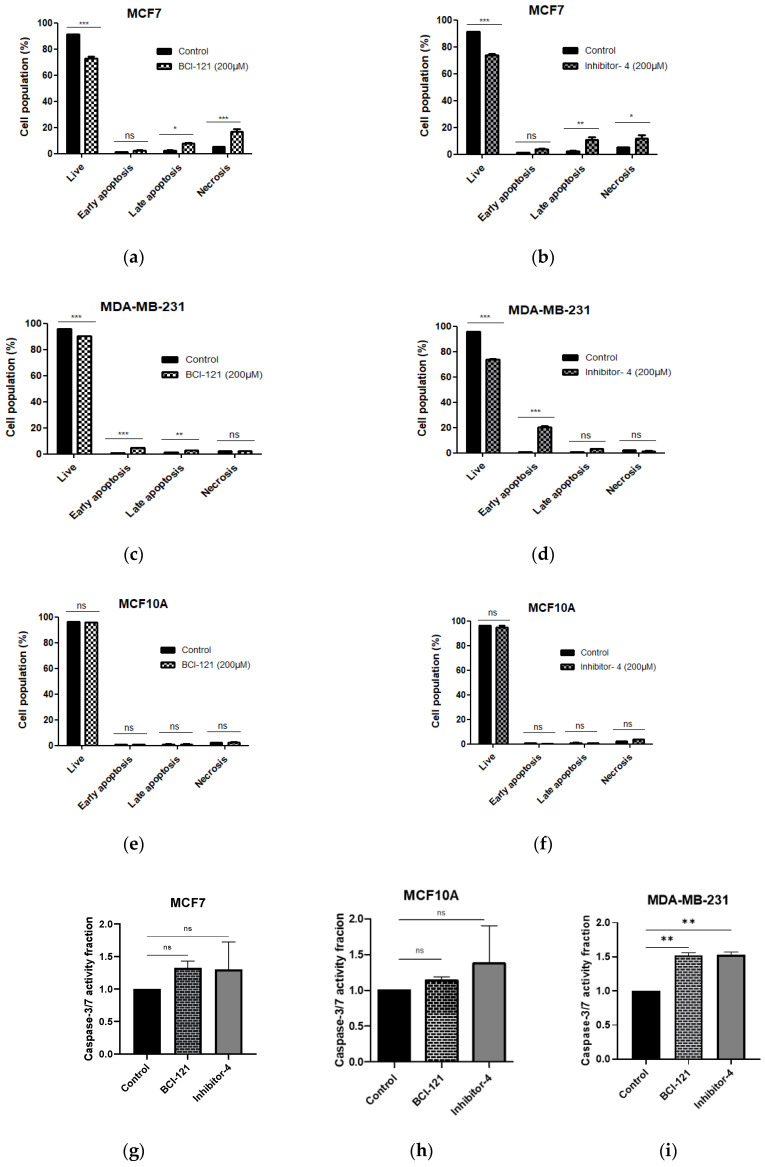
Cell apoptosis was assessed using APC Annexin V/PI and flow cytometry. (**a**,**b**) MCF7 (breast cancer), (**c**,**d**) MDA-MB-231 (breast cancer) and (**e**,**f**) MCF10A (normal breast) cell lines were treated with SMYD3 inhibitors for 48 h. (**a**,**c**,**e**) Apoptosis after 48 h of BCI-121 treatment on each cell line. (**b**,**d**,**f**) Apoptosis after 48 h of Inhibitor-4 treatment on each cell line. (**g**–**i**) Cell apoptosis was investigated with SMYD3 inhibitors using Caspase-3/7 activity assay. Values are mean ± standard error of the means. statistically significant differences from control are indicated by * *p* < 0.05, ** *p* < 0.01, *** *p* < 0.001 or ns *p* > 0.05.

**Table 1 ijms-21-09549-t001:** Summary of SMYD3 availability and activity in the cell lines used in this study.

Cancer Cell Lines	Origin	SMYD3 Expression	Assay	Methylation Activity	References
MCF7/MDA-MB-231	Human epithelial breast cancer cells	High	Western blot, RT-qPCR	H4K5, H3K4	[8,14,18,19,20]
MCF10A	Human epithelial breast cells	Very low	Western blot, RT-qPCR	H4K5, H3K4	[18,19]

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
