# Peer review of "In Silico/In Vitro Hit-to-Lead Methodology Yields SMYD3 Inhibitor That Eliminates Unrestrained Proliferation of Breast Carcinoma Cells"

_ijms, 2020, doi:10.3390/ijms21249549_

Round 1
Reviewer 1 Report
SMYD3 is overexpressed also in MDA-MB231, a triple negative breast Cancer line. Could you improve the correlation among ER, AR and SMYD3?
You can see the following manuscripts
https://doi.org/10.3389/fendo.2018.00492
https://doi.org/10.4252/wjsc.v11.i9.594
Experiments should be performed also in MDA-MB231.
Please specify the cell type only the first time that you mention it.
How do you explain that a normal cell line show a lower doubling time than a cancer cell line?
A western blot analysis of proteins involve ed in apoptosis could be useful.
Author Response
Responses to Reviewer 1:
- SMYD3 is overexpressed also in MDA-MB231, a triple negative breast Cancer line. Could you improve the correlation among ER, AR and SMYD3?
We agree that this improves the coherence of our paper. The correlation between SMYD3 and ER was added in the Introduction (page 2, lines 62-65)
- Experiments should be performed also in MDA-MB231.
Good catch. We have conducted this experiment and added to Figure 3 (page 5), and
Results page 5 at lines 148-150.
- Please specify the cell type only the first time that you mention it.
The requested change has been applied throughout the manuscript unless there are differences in the results between the two breast cancer cell lines that requires to specify the cell type.
- How do you explain that a normal cell line showed a lower doubling time than a cancer cell line?
The cell lines were purchased and grown as recommended by ATCC. During the characterization of cell lines, the doubling time from collected data was compatible with ATCC documents. Each cell line has a different doubling time that can be different from other cell lines. This difference can be explained by the different growth medium and supplements added for each. Furthermore, cancer cells do not necessarily proliferate faster than normal cell lines; rather cancer cells are typically characterized by unceasing proliferation. Given the fact that the effect of the SMYD3 inhibitor on each cell line was done separately, the doubling time can be different for each cell line (e.g. compared MCF7 baseline to time points after addition of the inhibitor after 24 h and 48 h)
- A western blot analysis of proteins involved in apoptosis could be useful.
Because of the time limitation and COVID19 situation we performed an assay measuring Caspase-3/7 activity instead of western blot. Supportive data were included in the manuscript (Figure 5 page 8, Results at line 238 and Methods at line 447) Hopefully, conducting this alternative method will be acceptable.

Reviewer 2 Report
Within the manuscript “In silico/in vitro hit-to-lead methodology yields SMYD3 inhibitor that eliminates unrestrained proliferation of breast carcinoma cells” Ilham M. Alshiraihi and colleagues implement the Small Molecule Drug Discovery Suite from Schrodinger to determine and uncover new potential SMYD3 inhibitors. If 3D structures of proteins are available this software can provide a list of candidate purchasable compounds. A tremendous advantage is the access to possible compounds to any laboratory without the need to develop and/or test libraries, with the limitation of availability of the 3D structure. Using this method, Ilham M. Alshiraihi and colleagues test five potential SMYD3 inhibitors and further test one showing efficacy against the in vitro methyltransferase activity of SMYD3. This compound is tested in breast cancer and normal cell lines that, based on the literature, overexpress or not SMYD3. They report some efficacy of this new SMYD3 inhibitor in reducing proliferation and inducing apoptosis of breast cancer cells. This new compound, if specific, could be very useful to inhibit SMYD3 in future studies and to further test potential therapeutic strategies.
Major comments
- The authors choose their cell lines models based on published literature and use breast cancer cell lines overexpressing SMYD3 (MCF7 and MDA-MB-231) and one “normal” breast cell line not presenting SMYD3 overexpression as a control. However, the authors have not tested if the cell lines they purchased from ATCC express SMYD3 as expected. A western blot looking at SMYD3 protein expression in these cells would be necessary to establish and confirm this choice of cell lines.
- Throughout the manuscript the authors establish that Inhibitor-4 has much less effect than the positive control inhibitor on normal cells and conclude that Inhibitor-4 is more selective. The fact that this inhibitor does not affect the proliferation etc of normal cells is not a proof of selectivity. In these lines, the authors should perform experiments assessing such selectivity: how do they know this inhibitor does not affect other SMYD family members, if not other proteins at large? One suggestion would be to perform SMYD3 knock-down in parallel to inhibitor treatment and compare gene expression.
- Why are the authors not testing MDA-MB-231 in Figure 2?
Minor comments
- Stating what BCI-121 is in the main text (in addition to the method section) would help the reader in section 2.2
Author Response
- The authors choose their cell lines models based on published literature and use breast cancer cell lines overexpressing SMYD3 (MCF7 and MDA-MB-231) and one “normal” breast cell line not presenting SMYD3 overexpression as a control. However, the authors have not tested if the cell lines they purchased from ATCC express SMYD3 as expected. A western blot looking at SMYD3 protein expression in these cells would be necessary to establish and confirm this choice of cell lines.
Western blot and immunocytochemistry were performed to confirm the overexpression of SMYD3 in the cancer cell line and supporting figures and data have been included in Figure 2 (page 4), Results (line 132), and Methods (lines 367 and 380).
- Throughout the manuscript the authors establish that Inhibitor-4 has much less effect than the positive control inhibitor on normal cells and conclude that Inhibitor-4 is more selective. The fact that this inhibitor does not affect the proliferation etc of normal cells is not a proof of selectivity. In these lines, the authors should perform experiments assessing such selectivity: how do they know this inhibitor does not affect other SMYD family members, if not other proteins at large? One suggestion would be to perform SMYD3 knock-down in parallel to inhibitor treatment and compare gene expression.
We agree that a SMYD3 knockdown experiment would be a thorough way to assess inhibitor selectivity. However, performing a knockdown experiment in the provided time for revision was not possible, however we will certainly consider this experiment in our future work and characterization.
- Why are the authors not testing MDA-MB-231 in Figure 2?
This is a good catch. We have now tested MDA-MB-231 cells and the supporting data were added in Figure 3 (page 5) and Results (line 148).
- Stating what BCI-121 is in the main text (in addition to the method section) would help the reader in section 2.2
We have added a description of BCI-121 at the end of the introduction and included its relevant reference (line 87). We agree that it should not be introduced in the results section 2.2 without any previous reference.

Round 2
Reviewer 1 Report
Authors addressed my concerns. I accettato the manuscript in the present form.
Reviewer 2 Report
Within the revised version of “In silico/in vitro hit-to-lead methodology yields SMYD3 inhibitor that eliminates unrestrained proliferation of breast carcinoma cells” Ilham M. Alshiraihi and colleagues did not address one of my major concern:
Throughout the manuscript the authors establish that Inhibitor-4 has much less effect than the positive control inhibitor on normal cells and conclude that Inhibitor-4 is more selective. The fact that this inhibitor does not affect the proliferation etc of normal cells is not a proof of selectivity. In these lines, the authors should perform experiments assessing such selectivity: how do they know this inhibitor does not affect other SMYD family members, if not other proteins at large? One suggestion would be to perform SMYD3 knock-down in parallel to inhibitor treatment and compare gene expression.
Response from the authors: We agree that a SMYD3 knockdown experiment would be a thorough way to assess inhibitor selectivity. However, performing a knockdown experiment in the provided time for revision was not possible, however we will certainly consider this experiment in our future work and characterization.
I understand the authors did not have the time to address this concern, also taking into account the limitations laboratories can face with the pandemic situation. However, if this point is not addressed the authors cannot conclude that their inhibitor specifically inhibits SMYD3 activity in cells. In that regard they need to change the text accordingly and add this caveat in the discussion/conclusion section.
Lane 155: suggests instead of demonstrates. Lane 192: suggests instead of indicates. Lane 296: remove “specifically” as this is not shown. Conclusion: do not conclude that the inhibitor is specific. Specificity is not shown and a discussion related to this needs to be added in the discussion section, as important follow-up experiments for example.
